Determinants of pro-environmental behavior among excessive smartphone usage children and moderate smartphone usage children in Taiwan

Fang Wei-Ta 1
Ng Eric 2
Liu Shu-Mei liushumei@hznu.edu.cn shumei80405010e@gmail.com 3
Chiang Yi-Te 1
Chang Mei-Chuan 1
1 Graduate Institute of Environmental Education, National Taiwan Normal University , Taipei , Taiwan
2 School of Management and Enterprise, University of Southern Queensland , Queensland , Australia
3 Department of Preschool Education, College of Education, Hangzhou Normal University , Hangzhou , China
Borghi Anna
Electronic publication date: 2021 Jun 18
Publication date: 2021
Volume: 9
Electronic Location ID: e11635
Received 2020 Oct 29; Accepted 2021 May 27
Copyright: ©2021 Fang et al.
Copyright year: 2021
Copyright holder: Fang et al.
License: This is an open access article distributed under the terms of the Creative Commons Attribution License, which permits unrestricted use, distribution, reproduction and adaptation in any medium and for any purpose provided that it is properly attributed. For attribution, the original author(s), title, publication source (PeerJ) and either DOI or URL of the article must be cited.
License URL: https://creativecommons.org/licenses/by/4.0/

Keywords: Pro-environmental behavior, Perceived behavioral control, Personal norms, Social norms, Smartphone usage, Children

Funding: Ministry of Science and Technology MOST 105-2511-S-003-021-MY3 National Taiwan Normal University (NTNU), Taiwan, ROC This research was funded by the Ministry of Science and Technology (grant number MOST 105-2511-S-003-021-MY3); and subsidized by the National Taiwan Normal University (NTNU), Taiwan, ROC. The funders had no role in study design, data collection and analysis, decision to publish, or preparation of the manuscript.

==============================
Introduction

Although there is evidence linking the relationships between smartphone usage with health, stress, and academic performance, there is still inadequate knowledge about the influence on pro-environmental behaviors. This study seeks to bridge this gap by adapting the theory of attribution framework to examine the effects of personal norms, social norms, perceived behavioral control on pro-environmental behavior of smartphone usage in children.

Methods

A total of 225 children aged between 11 to 12 from eight selected public primary schools at the Hsinchu Science and Industrial Park in Taiwan were surveyed. Two distinct groups (excessive versus moderate usage) were purposefully selected for comparison, of which 96 participants were excessive smartphone users while the remaining 129 were moderate smartphone users.

Results

Findings revealed significant differences between excessive and moderate smartphone usage children groups in personal norms (p < 0.001), social norms (p = 0.002), perceived behavioral control (p = 0.001), and pro-environmental behavior (p = 0.001). Findings for excessive smartphone usage children showed that social norms (β = 0.428, t = 4.096***, p < 0.001) had a direct predictive impact on pro-environmental behavior. In contrast, while there was no direct path established between personal norms and pro-environmental behavior (β = 0.177, t = 1.580, p > 0.05), as well as social norms and pro-environmental behavior for moderate smartphone usage children (β = 0.181, t = 1.924, p > 0.05), but such a relationship could be developed through the mediating effect of perceived behavioral control (β = 0.497, t = 4.471***, p < 0.001).

Discussion

The results suggested that excessive smartphone usage children lack positive perceived behavioral control, and their pro-environmental behavior could only be predicted through explicit social norms, whereas pro-environmental behavior of moderate smartphone usage children was implicitly influenced by personal norms through perceived behavioral control.

Introduction

Nowadays, smartphones are becoming increasingly popular that have brought many changes to our day-to-day lives, particularly with the ease of access to a vast variety of mobile applications for the purpose of internet browsing, gaming, social networking, communication, and so on. This phenomenon has seen the number of smartphone users grow steadily from 2.5 billion in 2016 to 2.9 billion in 2018 and is expected to reach 3.8 billion by 2021 globally (Statista, 2020). The growth is stimulated by the many benefits (e.g., entertainment, banking, socializing, and gaming) offer by smartphones, and this has been attested by past studies (Kang & Jung, 2014; Lemola et al., 2015; Susanto, Chang & Ha, 2016). Although there are numerous advantages of using smartphones, but several side effects have also arisen due to the excessive smartphone usage. In particular, the widespread use of smartphones is regarded as a factor influencing pro-environmental behavior, leading to disconnect people from the natural environment which has become substantial worldwide psychological and behavioral issues (Kadir, Mehmet & Abdullah, 2015; Lee et al., 2014; Chiang et al., 2019). While it remains controversial whether smartphone usage cause disconnectedness between people and natural environment (Fletcher, 2017; Miles, Zaheer & Mark, 2018), studies have found that adults who spent extended amount of time on smartphone in a day are more likely to exhibit a stronger negative pro-environmental behavior (Kesebir & Kesebir, 2017; Miles, Zaheer & Mark, 2018). Such extensive smartphone usage behavior is not only limited to adults, but also evident in children who are increasingly smartphone users nowadays. Smartphone technology has changed the growth and development of children, and there is evidence to support a close correlation between health, behavior, and smartphone usage (Dennison et al., 2013; Lee et al., 2014).

Studies show that children have spent more time and energy on smartphones nowadays that have led to a decrease in their physical outdoor activities and curiosity about the natural environment (Haug et al., 2015; Tian et al., 2018). Other studies have also investigated several different aspects of smartphone usage, such as factors influencing smartphone usage (Aljomaa et al., 2016; Park et al., 2013), impact of smartphone usage on social relationships (Choi, Lee & Ha, 2012), the effects of smartphone usage on academic performance (Samaha & Hawi, 2016), the relationship between smartphone usage and stress (Chiu, 2014; Wang et al., 2015), and relationship between smartphone usage and social anxiety and loneliness (Gao et al., 2016). However, there are very few studies attempted to combine the two fields of smartphone usage and environmental behaviors together, let alone explore their relationship. Therefore, the key focus of this study is to fill this gap by specifically investigating the impact of smartphone usage on children’s pro-environmental behaviors. This is particularly important because experiences gained through interactions with the environment during the early human development phase can have a considerable influence on a person’s perception of the environment (Bandura, 2006).

Conceptual framework and hypotheses

In the present study, we used Attribution Theory to explain the causes of children’s pro-environmental behavior. Pro-environmental behavior refers to the behavior of a person who consciously minimizing his/her negative impact on the environment (Kollmuss & Agyeman, 2002). Attribution Theory posits that a behavior can be influenced by situations that arise from internal and/or external attributions (Heider, 2013; Kassin & Fein, 2010). Internal attributions refer to causes of behavior related to some internal characteristic, and two key elements involve are: (1) perceived behavioral control, and (2) personal norms (Onu, Oats & Kirchler, 2019; Wated & Sanchez, 2005; Xu et al., 2020). Perceived behavioral control is related to an individual’s perceived ease or difficulty of performing personal capabilities to control external challenges (Ajzen, 1985). Whereas personal norms refer to the internal motivation of an individual’s perceived moral obligations when taking an action; it is a kind of environmental self-awareness and self-discipline, which is believed to be associated with the generation of pro-environmental behavior (Ajzen & Driver, 1991; Bertoldo & Castro, 2016; Esfandiar, Pearce & Dowling, 2019; Thøgersen, 2006). External attributions are causes of behavior that derive from situations outside a person’s control. Social norms are a key component of external attributions that goes beyond attitudes that shape people’s behaviors with an attempt to conform to a perceived norm (Weiner, 2001; Yoon & Lee, 2016). Social norms refer to what people generally believe to be typical behaviors or actions in the group and there are reciprocal expectations of the people within a reference group (Paluck et al., 2010). Such an inter-dependence of expectation and action can often result in a strong resistant to change by people. The relationship and influence of these three key variables (i.e., perceived behavioral control, personal norms, social norms) on pro-environmental behaviors will be further discussed next.

Personal norms, social norms and pro-environmental behaviors

Studies using the theory of value-belief-norm have found personal norms to be one of the key motivations for individuals to be more autonomous and self-demanding to adopt pro-environmental behaviors (Abrahamse et al., 2009; Stern et al., 1999). In a study by Quinn & Burbach (2008), farmers’ pro-environmental behaviors to improve surface water quality are found to be positively influenced by personal norms. Other previous studies have also suggested that pro-environmental behaviors are positively affected by personal norms. For example, personal norms can be used to predict recycling and environmentally friendly consumer behaviors (Ahn, Koo & Chang, 2012; Thøgersen, 2006; Turaga, Howarth & Borsuk, 2010), influence conservation behaviors (Kaiser, Hübner & Bogner, 2005), increase riding on public transportation (Thøgersen, 2006), and reduce car usage (Abrahamse et al., 2009). Therefore, to understand the effect of personal norms on pro-environmental behaviors of children using smartphone in Taiwan, the following hypotheses are proposed:

Hypothesis 1 (H1). Personal norms influence the pro-environmental behavior of excessive smartphone usage children.

Hypothesis 6 (H6). Personal norms influence the pro-environmental behavior of moderate smartphone usage children.

Farrow, Grolleau & Ibanez (2017) have identified various conceptualizations of social norms that are used to investigate the effects of social norms on pro-environmental behavior, and there is evidence to support the relationship. Previous studies have also used different theoretical models of behavior (e.g., Theory of Planned Behavior, Theory of Normative Expectations) from the social psychology and economics literature to explore the influence of social norms on pro-environmental behaviors, and the findings indicate the existence of a relationship between them (Azjen, 1991; Lapinski & Rimal, 2005). Vu et al. (2020) reveal that peer farmers’ behaviors have a positive influence on farmers’ adoption of organic fertilizer, and socially desired behaviors have been used more broadly to encourage pro-environmental farming practices in developed countries. Findings from a study by Duron-Ramos et al. (2020) suggest that children who show more altruistic values and sociable behavior toward others are more likely to demonstrate pro-environmental behaviors. Other studies have also found social norms affect recycling, organic food purchases, and the utilization of public transportation behaviors (Ferdinando et al., 2011; Thøgersen, 2006). Thus, this study seeks to understand the role that social norms play in the pro-environmental behaviors of children using smartphone in Taiwan. Given the above, two hypotheses are proposed:

Hypothesis 2 (H2). Social norms influence the pro-environmental behavior of excessive smartphone usage children.

Hypothesis 7 (H7). Social norms influence the pro-environmental behavior of moderate smartphone usage children.

Personal norms, social norms and perceived behavioral control

Perceived behavioral control can be divided into internal and external controls. Internal control factors relate to personal behaviors such as personal skills, capabilities, or emotions, while external control factors include information, opportunities, and dependence on others from the outside world (Chan & Bishop, 2013). Thus, perceived behavioral control is likely to be affected by both internal (i.e., personal norms) and external (i.e., social norms) attributions.

Previous studies related to reduce car usage (Abrahamse et al., 2009), adoption of organic fertilizer (Vu et al., 2020), energy saving intention (Ru, Wang & Yan, 2018), sustainable food consumption (Han & Hansen, 2012), and binning behavior in national parks (Esfandiar, Pearce & Dowling, 2019) have shown that personal norms affect perceived behavioral control. A study by Wall, Devine-Wright & Mill (2008) on commuter travel-mode choices indicate that personal norms and perceived behavioral control are invoked together and can have significant influence on car usage intention. Although personal norms may be present, but a low level of perceived behavioral control can still prevent that norm from being converted into pro-environmental behaviors. Thus, it is proposed that the following hypotheses are examined in relation to the pro-environmental behaviors of children using smartphone in Taiwan.

Hypothesis 3 (H3). Personal norms influence the perceived behavioral control of excessive smartphone usage children.

Hypothesis 8 (H8). Personal norms influence the perceived behavioral control of moderate smartphone usage children.

Prior studies have also revealed that social norms are related to perceived behavioral control (Bamberg & Möser, 2007). In the context of recycling and organic food purchase behaviors, Ferdinando et al. (2011) suggest that social norms have positive influence on perceived behavioral control. Another study by Leung & Rosenthal (2019) also found that there is a link between social norms and perceived behavioral control toward recycling intention and behaviors. When fellow colleagues are supportive of a particular behavior, individuals tend to perceive a stronger norm and have more behavioral control that contribute to positive recycling behaviors. Other areas of studies that suggested a relationship between social norms and perceived behavioral controls include: waste separation (Zhang et al., 2015), adolescents’ environmental intention (Lee, 2011), green poultry by farmers (Gholamrezai, Aliabadi & Ataei, 2021), tourists’ responsible environmental behaviors (Wang et al., 2018). As such, the following hypotheses are proposed to explore the pro-environmental behaviors of children using smartphone in Taiwan.

Hypothesis 4 (H4). Social norms influence the perceived behavioral control of excessive smartphone usage children.

Hypothesis 9 (H9). Social norms influence the perceived behavioral control of moderate smartphone usage children.

Perceived behavioral control and pro-environmental behavior

According to Ajzen (1991), perceived behavioral control is a crucial factor in the Theory of Planned Behaviors that affects behaviors. Past studies in the fields of conservation behavior intention (Kaiser, Hübner & Bogner, 2005), household-waste prevention behaviors (Bortoleto, Kurisu & Hanaki, 2012), air pollution prevention behaviors (Liu et al., 2018), recycling behaviors (Leung & Rosenthal, 2019), and eco-friendly behaviors of travelers (Han, 2015) have shown that perceived behavioral control plays a role in influencing pro-environmental behaviors. Given that perceived behavioral control may be influenced by both personal and social norms, it is posit that perceived behavioral control can possibly be a moderator between personal norms, social norms, and pro-environmental behaviors. Thus, the following hypotheses are proposed for the pro-environmental behaviors of children using smartphone in Taiwan.

Hypothesis 5 (H5). Perceived behavioral control influences the pro-environmental behavior of excessive smartphone usage children.

Hypothesis 10 (H10). Perceived behavioral control influences the pro-environmental behavior of moderate smartphone usage children.

This study aims to investigate the pro-environmental behavior of children in Taiwan, whereby the ownership of smartphones in children aged between 10 to 12 has grown significantly from 38.7% in 2013 to 82.7% in 2019 (The Child Welfare League Foundation, 2019b). This alarming trend has elicited considerable related discussions (e.g., counter measurements for smartphone usage, responsibilities of the government and parents, social impact, cyber bullying), and is an increasingly concerning social phenomenon in Taiwan that requires further understanding of the situation (The Child Welfare League Foundation, 2019a). This research seeks to examine how internal (i.e., perceived behavioral control, personal norms) and external (i.e., social norms) attributions are related to children’s pro-environmental behavior. Thus, this study proposed the following research framework (Fig. 1) to be investigated.

Figure 1 Proposed research framework.

Social norms is a key component of external attributions that goes beyond attitudes that shape people’s behaviors, and can be further classified as (1) subjective norms, and (2) descriptive norms.

Materials & Methods

Research area

This study was carried out in Hsinchu City, Taiwan, one of the major technological and industrial cities in Asia as described in Fang, Ng & Chang (2017). Specifically, the research was undertaken at the elementary schools around the Hsinchu Science Park, which was founded in 1980 with a development area of 653 hectares at the Hsinchu Park that resembles the Silicon Valley in the United States (Hsinchu Science Park, 2016). In 2017, there were 520 companies in Hsinchu Science Park with a total of 150,000 employees that generated an overall revenue exceeding NT$1 trillion (Hsinchu Science Park, 2016). There were several elementary schools situated in the Hsinchu Science Park area that cater to the needs of many families working there.

Participants and procedure

A probability cluster sampling method was adopted in this study with the selection of eight public elementary schools around the Hsinchu Science and Industrial Park located at Hsinchu City, Taiwan as described in Fang, Ng & Chang (2017). A total of 16 classes (one each from grades five and six, age between 11 to 12) were randomly selected from eight elementary schools. Prior to conducting this research study, written information (e.g., purpose and objectives of the study, no financial incentives or payments involve) about the survey and consent forms were sent to children, parents, school principals and class teachers to brief them about the research and obtain their agreement to participate in the study as outlined in Fang, Ng & Chang (2017). At the same time, their rights to withdraw from the study at any time were also conveyed to them. Participants were again informed verbally about the purpose of the study and procedures to follow, as well as given the opportunity to have their queries answered prior to completing the survey. Participants had approximately 15 min in class to complete the questionnaire survey and the class teacher would collate them when completed as described in Fang, Ng & Chang (2017). All participants, parents, class teachers and school principals provided written informed consent. The National Taiwan Normal University Research Ethics Committee had approved (201707HS001) this study and endorsed informed consent acquired from the respective class teachers and school principals as adequate. In addition, parents could opt their child out of the study at any time prior to the questionnaire commencing. A total of 260 questionnaires were distributed and were all returned. However, 35 were considered invalid due to their incomplete responses. Therefore, statistical analysis was performed using the remaining 225 questionnaires.

Measures

This study was supported by the literature reviewed on the impact of cognitive and psychological factors on pro-environmental behavior (De Leeuw & Valois, 2014; De Leeuw et al., 2015; Esfandiar, Pearce & Dowling, 2019). Data were collected as previously described in Fang, Ng & Chang (2017). As described in Fang, Ng & Chang (2017), this study specifically sought to measure three key dimensions: (1) perceived behavioral control, (2) personal norms, and (3) social norms that were considered influential to pro-environmental behavior (Culiberg & Elgaaied-Gambier, 2016; Esfandiar, Pearce & Dowling, 2019; Farrow, Grolleau & Ibanez, 2017). This research adopted the self-administered questionnaire survey method that included two key sections related to background and psychological variables (please see Appendix A1).

Background variables

In the background variables, personal data were gathered about participants’ gender, grade level, and smartphone usage. Gender and grade level were measured through nominal variables (i.e., Gender –Male or Female; Grade level –Grade 5 or Grade 6), whereas smartphone usage was measured by the question “How many hours per day do you usually spend on smartphone during a typical day?” with five possible multiple-choice responses (i.e., none, less than 1 h, between 1 to 2 h, between 2 to 3 h, between 3 to 4 h, between 4 to 5 h, more than 5 h). While there is no one single agreed definition for excessive smartphone usage, but it can be broadly defined as the extended use of smartphone by individuals that interferes their daily lives (Hair Jr et al., 2016). Studies have shown that there were about 50% of children used smartphones for more than 2 h per day (Haug et al., 2015), and approximately 70% of children’s daily screen time exceeded the American Academy of Pediatrics recommendations (i.e., more than two hours per day) (Hair Jr et al., 2014; Henseler, 2017). Thus, for the purpose of this study, excessive smartphone usage is considered to be more than two hours per day. In this research study, two groups of children (i.e., excessive smartphone usage group - more than two hours per day of smartphone usage, and moderate smartphone usage group - less than 2 h of smartphone usage per day) were purposefully selected for comparison.

Psychological variables

The psychological variables in the questionnaire survey were mainly compiled and adapted from previously conducted pro-environmental behavior research studies, specifically related to perceived behavioral control, social norms and personal norms dimensions (Abrahamse et al., 2009; Henseler, 2017; Thøgersen, 2006). As for the questionnaire items for pro-environmental behavior, they were predominantly derived from the classification by Erdogan & Ok (2011).

Three experts and 50 children in grades five and six had pre-tested the questionnaire survey and some changes (e.g., simplified wordings) had been made accordingly to ensure an easier understanding of the context. Nonetheless, the original meaning of the items adapted from the previous studies were retained. A five-point Likert scale (i.e., 1 = “Strongly disagree” to 5 = “Strongly agree”) was applied for the measurement of the personal norms, social norms, perceived behavioral control, and pro-environmental behaviors in this study as described in Fang, Ng & Chang (2017). The Cronbach’s α values of the respective dimensions were perceived behavioral control (0.686), personal norm (0.766), social norm (0.886), and pro-environmental behavior (0.846), which demonstrated internal reliability since their values were greater than the 0.6 requirement. The Kaiser–Meyer–Olkin and Spherical Bartlett tests as described in Fang, Ng & Zhan (2018) had indicated a value of 0.885 and 2496.758 (p < 0.001) respectively. Factor analyses were subsequently conducted in PLS and the factor loading of each question in the key dimensions had exceeded the value of 0.4 (Pett, Lackey & Sullivan, 2003). Given the outcome of these test values, the scales used for the psychological variables could be regarded as reliable.

Statistical analysis

The statistical analysis in this study was carried out as previously described in Fang, Ng & Chang (2017) and Fang, Ng & Zhan (2018), using the Statistical Package for Social Sciences (version 22) software program. Specifically, frequency analysis was used to identify the total number of occurrences, the mean and standard deviation (SD) scores for the demographic questions and items in the key dimensions of perceived behavioral control, personal norms, social norms, and pro-environmental behavior. In addition, Pearson’s correlation coefficient was used to evaluate the intensity and direction of the relationship between the key dimensions. Whereas the multiple regression analysis was used to determine the influence of perceived behavioral control, personal norms, social norms on pro-environmental behavior. The SmartPLS 2.0 statistical software was utilize in this study to perform the path analysis, which were then used to assess the magnitude and significance of the causal relationships between perceived behavioral control, personal norms, social norms, and pro-environmental behavior.

Results

Descriptive statistics

Overall results showed that both males (48.9%, n = 110) and females (51.1%, n = 115) were relatively well represented in this study. Among them, 47.6% (n = 107) were in grade five and the remaining 52.4% (n = 118) were in grade six. Of the two control groups investigated, the excessive smartphone usage group accounted for 42.7% (n = 96), whereas the moderate smartphone usage group was 57.3% (n = 129). Findings revealed significant differences between males and females in their perceived behavioral control (df = 223, two-tailed, t = 2.126 > 1.96, p = 0.035). Similarly, significant differences were also identified between grade five and grade six children in terms of their personal norms (df = 223, two-tailed, t = 2.095 > 1.96, p = 0.037) and perceived behavioral control (df = 223, two-tailed, t = 2.498 > 1.96, p = 0.013). Furthermore, significant differences between excessive smartphone usage group children and moderate smartphone usage group children were also evident in personal norms (df = 223, two-tailed, t = 4.693 > 1.96, p < 0.001), social norms (df = 223, two-tailed, t = 3.205 > 1.96, p = 0.002), perceived behavioral control (df = 223, two tailed, t = 3.465 > 1.96, p = 0.001), and pro-environmental behavior (df = 223, two-tailed, t = 3.520 > 1.96, p = 0.001). Table 1 presents a brief summary of the demographic findings and their association with personal norms, social norms, perceived behavioral control, and pro-environmental behavior.

As shown in Table 2, the mean score for the overall perceived behavioral control was 3.62. Among the four perceived behavioral control related items, “I can save water resources” had the highest mean score (3.92), and this was followed by “I can observe environmental cleanliness” (3.80), “I take the initiative to go outdoors” (3.38), and “I have involvement in and disseminate information beneficial to the environment” (3.37). The results indicated an internal consistency reliability measurement with the Cronbach’s α value of 0.686 for perceived behavioral control related items.

Table 1 Descriptive statistics related to the demographic questions for personal norms, social norms, perceived behavioral control, and pro-environmental behavior.

Variables	Frequency	Percent (%)	Personal norms	Social norms	Perceived behavioral control	Pro-environmental behavior	
			Mean	SD	t	p	Mean	SD	t	p	Mean	SD	t	p	Mean	SD	t	p	
Gender																			
Female	115	51.1	3.76	0.791	1.665	0.097	3.60	0.78	1.161	0.247	3.73	0.78	2.126	0.035	3.43	0.96	1.768	0.078	
Male	110	48.9	3.59	0.777	3.48	0.78	3.50	0.84	3.21	0.91	
Grade level																			
Grade 6	118	52.4	3.58	0.76	−2.095	0.037	3.51	0.75	−0.655	0.513	3.49	0.81	−2.498	0.013	3.25	0.88	−1.101	0.272	
Grade 5	107	47.6	3.80	0.81	3.58	0.82	3.76	0.80	3.39	1.00	
Smartphone usage																			
Excessive	96	42.7	3.41	0.71	−4.693	0.000	3.43	0.68	−3.205	0.002	3.40	0.80	−3.465	0.001	3.07	0.90	−3.520	0.001	
Moderate	129	57.3	3.89	0.78	3.75	0.74	3.78	0.80	3.50	0.93	

Findings for the overall personal norms revealed a mean score of 3.68. There were three personal norms related items, namely “For self-discipline, I carry my own water cup when I go out” (3.79), “For self-discipline, I carry my own cutlery when I go out” (3.70), and “I must not arbitrarily abandon pets or plant garden plants in the wild” (3.45). Results of the personal norms related items are displayed in Table 3 below. There was a consistent reliable measurement for the personal norms related items with the Cronbach’s α value of 0.766.

Table 2 Descriptive statistics for perceived behavioral control related items.

The mean score for the overall perceived behavioral control was 3.62.

Perceived behavioral control	Mean	SD	
PBC1. I take the initiative to go outdoors.	3.38	1.16	
PBC2. I have involvement in and disseminate information beneficial to the environment.	3.37	1.05	
PBC3. I can save water resources.	3.92	1.07	
PBC4. I can observe environmental cleanliness.	3.80	0.96	
Overall perceived behavioral control	3.62	0.82	

Results (see Table 4) indicated that the mean score for the overall social norms was 3.61. There were six social norms related items whereby “People I know do not litter arbitrarily”, and “People I know go outdoors instead of staying indoors with air conditioning” had the highest (3.78) and lowest (3.22) mean score respectively. Other items also include: “People I know are aware of how to save water” (3.77), “People I know expect me to go outdoors and use less air conditioning” (3.68), “People I know want me to carry a water cup and cutlery when I go out” (3.63), and “People I know carry their own cutlery when they go out” (3.59). The Cronbach’s α value of 0.886 suggested internal consistency for the social norms related items.

Table 3 Descriptive statistics for personal norms related items.

There was a consistent reliable measurement for the personal norms related items with the Cronbach’s α value of 0.766.

Personal norms	Mean	SD	
PN1. For self-discipline, I carry my own cutlery when I go out.	3.70	1.08	
PN2. For self-discipline, I carry my own water cup when I go out.	3.79	1.06	
PN3. I must not arbitrarily abandon pets or plant garden plants in the wild.	3.45	1.08	
Overall personal norms	3.68	0.79	

In terms of the overall pro-environmental behavior, findings revealed a mean score of 3.32. There were five items related to pro-environmental behavior, and of which “I go outdoors in my free time, rather than watching TV and sitting in front of the computer” received the highest mean score (3.62). This was followed by “I participate in conducting surveys on animal and plant-related activities in the vicinity of the community” (3.52), “I persuade others to sort waste” (3.34), “I visit the park and volunteer” (3.14), and “I participate in environmental activities conducted outdoors” (2.97). Results of the pro-environmental behavior related items are outlined in Table 5 below. The internal consistency reliability for the pro-environmental behaviors related items was measured, with the Cronbach’s α value of 0.846.

Table 4 Descriptive statistics for social norms related items.

The results indicated that the mean score for the overall social norms was 3.61.

Social norms	Mean	SD	
SN1. People I know want me to carry a water cup and cutlery when I go out.	3.63	1.05	
SN2. People I know want me to go outdoors instead of staying indoors with air conditioning.	3.68	0.13	
SN3. People I know carry their own cutlery when they go out.	3.59	1.06	
SN4. People I know are aware of how to save water.	3.77	1.01	
SN5. People I know do not litter arbitrarily.	3.78	1.01	
SN6. People I know go outdoors instead of staying indoors with air conditioning.	3.22	1.22	
Overall social norms	3.61	0.79	

Table 5 Descriptive statistics for pro-environmental behavior related items.

The results of the pro-environmental behavior related items are outlined in the table. The internal consistency reliability for the pro-environmental behaviors related items was measured, with the Cronbach’s α value of 0.846.

Pro-environmental behavior	Mean	SD	
PEB1. I persuade others to sort waste.	3.34	1.211	
PEB2. I go into nature in my free time instead of using smartphone.	3.62	1.174	
PEB3. I participate in environmental activities conducted outdoors.	2.97	1.185	
PEB4. I participate in conducting surveys on animal and plant-related activities in the vicinity of the community.	3.52	1.203	
PEB5. I visit the park and volunteer.	3.14	1.200	
Overall pro-environmental behavior	3.32	0.940	

Correlation analysis

As shown in Table 6, the correlation analysis results indicated that personal norms and perceived behavioral control were highly correlated with a value of 0.723 (p < 0.01). The correlation coefficient of the remaining relationships indicated a moderate correlation; social norms and perceived behavioral control (r = 0.520, p < 0.01); perceived behavioral control and pro-environmental behavior (r = 0.598, p < 0.01); personal norms and pro-environmental behavior (r = 0.536, p < 0.01); social norms and pro-environmental behavior (r = 0.522, p < 0.01). Hence, personal norms, social norms, perceived behavioral control, and pro-environmental behavior were considered to be correlated.

Table 6 Pearson’s correlation matrix (Mean).

The correlation coeffcient of the remaining relationships indicated a moderate correlation; social norms and perceived behavioral control (r = 0.520, p < 0.01); perceived behavioral control and pro-environmental behavior (r = 0.598, p < 0.01); personal norms and pro- environmental behavior (r = 0.536, p < 0.01); social norms and pro-environmental behavior (r = 0.522, p < 0.01).

	Personal norms	Social norms	Perceived behavioral control	Pro-environmental behavior	
Personal norms	1.000				
Social norms	0.559	1.000			
Perceived behavioral control	0.723	0.520	1.000		
Pro-environmental behavior	0.536	0.522	0.598	1.000	
Notes.

All correlations are significant, p < 0.01 (two-tailed test).

Regression and path analysis

Results of the partial least squares regression analysis indicated different paths influencing pro-environmental behavior for both excessive smartphone usage, and moderate smartphone usage children. In the excessive smartphone usage group (as shown in Table 7), the average variance extracted (AVE) values for personal norms (0.5297), social norms (0.5363), perceived behavioral control (0.5219), and pro-environmental behaviors (0.5782) were above 0.50. The composite reliability (CR) values for personal norms (0.7660), social norms (0.8736), perceived behavioral control (0.8130), and pro-environmental behaviors (0.8711) were at least 0.70. Therefore, the model statistically supported the convergent validity and reliability for the measurement model between the latent variables and their respective dimensions (Hair Jr et al., 2016; Hair Jr et al., 2014; Henseler, 2017). The Cronbach’s α values for social norms (0.8280), and pro-environmental behaviors (0.8173) were above 0.7, and thus achieved a high level of internal consistency. While the Cronbach’s α values for personal norms (0.5604), and perceived behavioral control (0.6976) were below the value of 0.7, but they were higher than the acceptable value of 0.5, which indicated that the survey items had demonstrated an acceptable level of internal consistency. The explanatory power (R2) of the affected dimensions were perceived behavioral control (0.2865), and pro-environmental behavior (0.2964).

Table 7 Partial least square regression analysis of personal norms, social norms, and perceived behavioral control that predict pro-environmental behavior of excessive smartphone usage children (n= 96).

The results of the partial least squares regression analysis indicated different paths influencing pro-environmental behavior for both excessive smartphone usage, and moderate smartphone usage children.

	AVE	CR	R2	Cronbach’s α	
Personal norms	0.5297	0.7660		0.5604	
Social norms	0.5363	0.8736		0.8280	
Perceived behavioral control	0.5219	0.8130	0.2865	0.6976	
Pro-environmental behavior	0.5782	0.8711	0.2964	0.8173	

Figure 2 displayed the path analysis about the relationship between personal norms, social norms, perceived behavioral control, and pro-environmental behavior of excessive smartphone usage children. The bootstrapping method was used to obtain the t value of the path in order to examine the significant level. According to the path analysis for excessive smartphone usage children, personal norms (β = −0.044, t = 0.395, p > 0.05) had no direct influence on pro-environmental behavior, whereas social norms (β = 0.428, t = 4.096***, p < 0.001) had a direct predictive impact. Therefore, H1 was not supported as there was no evidence of a direct influence of personal norms on pro-environmental behavior for excessive smartphone usage children. However, H2 was supported since social norms had shown a direct positive influence on pro-environmental behavior for excessive smartphone usage children. Although personal norms (β = 0.304, t = 2.721**, p < 0.01), and social norms (β = 0.333, t = 2.779**, p < 0.01) had a significant impact on perceived behavioral control, but there was no evidence suggesting a relationship existed between perceived behavioral control (β = 0.220, t = 1.823, p > 0.05) and pro-environmental behavior. Hence, H3 and H4 were supported since both personal norms and social norms had a positive influence on perceived behavioral control for excessive smartphone usage children. Conversely, H5 was not supported because perceived behavioral control had no direct effect on pro-environmental behavior for excessive smartphone usage children.

Figure 2 Path diagram on personal norms (PN), social norms (SN), perceived behavioral control (PBC), and pro-environmental behavior (PEB) of excessive smartphone usage children.

The path analysis about the relationship between personal norms, social norms, perceived behavioral control, and pro-environmental behavior of excessive smartphone usage children is shown.

For the moderate smartphone usage children group (as shown in Table 8), the average variance extracted (AVE) values for personal norms (0.6549), social norms (0.6270), perceived behavioral control (0.6240), and pro-environmental behaviors (0.6441) had exceeded 0.50. The composite reliability (CR) values for personal norms (0.8496), social norms (0.9093), perceived behavioral control (0.8688), and pro-environmental behaviors (0.9004) were above 0.70. Thus, the model statistically supported the convergent validity and reliability for the measurement model between the latent variables and their respective dimensions (Hair Jr et al., 2016; Hair Jr et al., 2014; Henseler, 2017). The Cronbach’s α values for personal norms (0.7309), social norms (0.8794), perceived behavioral control (0.7989), and pro-environmental behaviors (0.8618) were greater than 0.7, and therefore indicated a high level of internal consistency. The explanatory power (R2) on the affected dimensions of perceived behavioral control, and pro-environmental behavior were recorded as 0.5697 and 0.5684 respectively.

Table 8 Partial least square regression analysis of personal norms, social norms, and perceived behavioral control that predict pro-environmental behavior of moderate smartphone usage children (n= 129).

All of the convergent validity, i.e. average variance extracted (AVE) values had exceeded 0.50, and similarly the composite reliability (CR) values were above 0.70.

	AVE	CR	R2	Cronbach’s α	
Personal norms	0.6549	0.8496		0.7309	
Social norms	0.6270	0.9093		0.8794	
Perceived behavioral control	0.6240	0.8688	0.5697	0.7989	
Pro-environmental behavior	0.6441	0.9004	0.5684	0.8618	

The path analysis (as shown in Fig. 3) displayed the relationship between personal norms, social norms, perceived behavioral control, and pro-environmental behavior of moderate smartphone usage children. In order to determine the level of significance, the bootstrapping method was used to obtain the t value of the path. Analysis results revealed that personal norms and social norms did not have a direct path relationship to pro-environmental behavior, but instead displayed an influence on perceived behavioral control. Therefore, H6 and H7 were not supported since there was no evidence of personal norms and social norms having direct influence on pro-environmental behavior for moderate smartphone usage children. Accordingly, personal norms had a greater impact on perceived behavioral control (β = 0.664, t = 10.357***, p < 0.001) than social norms (β = 0.153, t = 2.160*, p < 0.05), thus H8 and H9 were supported which confirmed the influence of personal norms and social norms on perceived behavioral control for moderate smartphone usage children. Although there was no direct path established between personal norms and pro-environmental behavior, and social norms and pro-environmental behavior, but such a relationship could be developed through the mediating effect of perceived behavioral control (β = 0.497, t = 4.471***, p < 0.001). The P-values of the paths from personal norms to pro-environmental behavior, and social norms to pro-environmental behavior were greater than the significance levels (α = 0.05). Therefore, H10 was supported, whereby perceived behavioral control had a positive influence on pro-environmental behavior for moderate smartphone usage children.

Figure 3 Path diagram on personal norms (PN), social norms (SN), perceived behavioral control (PBC), and pro-environmental behavior (PEB) of moderate smartphone usage children.

The relationship between personal norms, social norms, perceived behavioral control, and pro-environmental behavior of moderate smartphone usage children is shown.

Discussion

This study adopted the Social Attribution Theory to explore the impact of personal norms, social norms, and perceived behavioral control on pro-environmental behaviors by two groups of children in Taiwan based on their daily smartphone usage (i.e., excessive versus moderate). The extant literature suggested that there were inadequate studies conducted to understand the relationship between smartphone usage and environmental behaviors, which this study sought to fill the gap by investigating personal norms, social norms, and perceived behavioral control to determine their specific path and level of influence on pro-environmental behaviors. The overall results indicated that moderate smartphone usage children had shown a greater level of pro-environmental behaviors than those children who were excessive smartphone users, and this finding was supported by previous studies (Kesebir & Kesebir, 2017; Richardson, Hussain & Griffiths, 2018). The results also revealed support and accepted six (i.e., H2, H3, H4, H8, H9 and H10) of the 10 hypotheses, indicating significant direct positive relationship.

Excessive smartphone usage children group

Results suggested that the relationship between personal norms and pro-environmental behaviors for excessive smartphone usage children was considered weak, and as such H1 was rejected. This outcome indicated that personal beliefs of the excessive smartphone usage children had minimal influence on their pro-environmental behaviors. The absence of this relationship could be explained through a possible lack of one’s desire to conform to pro-environmental behaviors and/or a lack of environmental self-awareness and self-discipline that could contribute to exhibiting pro-environmental behaviors.

Findings revealed the support for H2 since social norm was a major predictive variable of pro-environmental behaviors for excessive smartphone usage children. This implied that excessive smartphone usage children were more likely to display and engage in pro-environmental behavior when they felt under pressure to conform to the expectation and requirement of normative social influence. This could be explained by the cultural context whereby Confucianism, which is an East Asian ethical and philosophical system that emphasizes on social values, has strong culture roots in countries such as China, Korea, Japan, and Singapore (Chang & Chu, 2007; Westwood & Chan, 1995). This strong emphasis on social values formed the basis for a sense of belonging to the group, which could explain the results emphasizing a greater influence of social norms for excessive smartphone usage children on their pro-environmental behavior. Hence, pro-environmental behaviors were only predicted through social norms, whereby excessive smartphone usage children followed the rules that regulate their life from explicit standards for proper behaviors.

Analysis results also indicated that while both personal norms and social norms had a significant impact on perceived behavioral control (i.e., H3 and H4) on the excessive smartphone usage children, but a relationship between perceived behavioral control and pro-environmental behaviors was not established (i.e., H5). The rejection of this hypothesis (H5) could be due to a lack of individual’s beliefs about their personal capabilities to overcome challenges that impeded their exhibition of pro-environmental behaviors. Therefore, pro-environmental behaviors were only predicted through social norms, whereby excessive smartphone usage children followed the rules that regulate their life from explicit standards for proper behaviors.

Moderate smartphone usage children group

For the moderate smartphone usage children group, findings rejected H6 and H7 since there was no evidence supporting the direct influence of personal norms on pro-environmental behaviors, and social norms on pro-environmental behaviors. This suggested that the pro-environmental behaviors of the moderate smartphone usage children would neither directly affected solely by their perceived moral obligations toward environment, nor when pressured to conform because of external social influences.

On the other hand, there was evidence in the moderate smartphone usage children group that indicated a direct influence from personal norms to perceived behavioral control, and social norms to perceived behavioral control, hence H8 and H9 were supported. Although personal norms would have provided moderate smartphone usage children the perceived moral obligations to undertake pro-environmental behaviors, but it did not appear to have a direct impact, instead an indirect influence on pro-environmental behaviors was only achieved through perceived behavioral control. An explanation to this could be that the perceived moral obligations to the environment itself was not strong enough for moderate smartphone usage children to undertake pro-environmental behaviors, but through self-resilience from perceived behavioral control, pro-environmental behaviors were attained (Ali et al., 2010; Schwarzer & Warner, 2013). Studies (e.g., Song & Li, 2019; Tchetchik, Kaplan & Blass, 2021) showed that individuals with self-resilience attitudes were more likely to demonstrate and adapt positive behavioral change to their surroundings, perception of green living environments and more responsible toward environmental issues. Therefore, personal norms could only affect pro-environmental behaviors through the mediating variable of perceived behavioral control (Abrahamse et al., 2009; Esfandiar, Pearce & Dowling, 2019). Findings also suggested that perceived behavioral control was a major predictive variable affecting pro-environmental behaviors (i.e., H10) for moderate smartphone usage children. This result aligned with other studies (e.g., De Leeuw et al., 2015) that found perceived behavioral control having a significant role to play in determining pro-environmental behaviors.

Differences between two groups

There were two key differences in the findings between excessive and moderate smartphone usage children investigated in this study. Firstly, results indicated that social norms were considered a key influencing factor for excessive smartphone usage children’s pro-environmental behavior. Social norm had a very strong direct positive association with behavioral intention for excessive smartphone users (Lee et al., 2018), and behavioral intention was the most direct predictor of behavior (Glanz, Rimer & Viswanath, 2015). Thus, social requirements and social expectation were more likely to predict pro-environmental behaviors of excessive smartphone usage children than moderate smartphone usage children. This suggested that excessive smartphone usage children might rely on the role of socializing agents in the development of self-regulation, such as parents (i.e., low levels of parental restrictive mediation), teachers, and peers (Chang et al., 2019), and therefore were unable to develop their own thoughts, feelings, and actions toward the exhibition of pro-environmental behavior (Zimmerman, 2000). As such, their norms could not be internalized due to a lack of self-regulation, resulting in their “following the instructions of the teachers and other adults or acting based on observed actions of the peers” behavior. Therefore, their demonstrated pro-environmental behavior was likely to be coercive and conditional under physical and emotional duress that were not necessarily the intended actual behavior of their own. This supported previous studies (Gökçearslan et al., 2016; Jeong et al., 2016) conducted about predictors of self-regulation on smartphone usage.

Secondly, the research results revealed that personal norms of moderate smartphone usage children had a greater predictive power on their perceived behavioral control, which subsequently affected pro-environmental behaviors. This indicated that moderate smartphone usage children had internalized their personal norms, and with the perceived ease of their capabilities, to perform pro-environmental behaviors. Hence, the personal norms of the moderate smartphone usage children could have an indirect positive impact on their pro-environmental behaviors via perceived behavioral control. The findings align with previous studies (e.g., Turaga, Howarth & Borsuk, 2010; Thøgersen, 2006; Abrahamse et al., 2009) conducted on the effects of personal norms on environmental behaviors.

Implications, limitations and future research

There are two key implications from the findings in this study. Firstly, pro-environmental behavior of excessive smartphone usage children was predominantly affected by social norms only. Therefore, it is recommended that environmental education programs (formal and informal) at schools need to be targeting independent of knowledge so that children can better understand the relationship between their behavior and the environment. This helps toward attaining a more sustainable pro-environmental behavior of becoming a socially responsible citizen and not just conforming because of the social norm pressure. Next, the increasing concern of excessive smartphone usage by children has also prompted the need to explore different ways to minimize such a harmful behavior. Through the promotion of more outdoor activities and utilizing outdoor venues such as parks and playgrounds, children are then provided with more choices other than engaging with their smartphones (e.g., gaming, internet surfing), therefore reducing the potential of smartphone overuse.

There are several limitations in this research study. First, the primary schools were selected in the proximity of the Hsinchu Science and Industrial Park location and thus the findings are not applicable to other geographical areas. To generalize the findings, further research needs to be conducted with a larger sample and tested accordingly. Secondly, research can also be investigated in other country contexts in the future to provide cross-country comparisons. Lastly, follow-up studies can be conducted to gain further in-depth understanding about the relationship of smartphone usage, and other psychological variables (such as attitudes, beliefs) toward environmental knowledge and behaviors.

Conclusions

This study has used the Attribution Theory as the basis to examine the causes of children’s pro-environmental behavior in Taiwan. Research results revealed that social norms were the key factor affecting the pro-environmental behavior of excessive smartphone usage children. Conversely, personal norms, through the indirect path of perceived behavioral control, had the greatest level of influence on moderate smartphone usage children’s pro-environmental behavior. The findings clearly distinguished the pro-environmental behavior path modes between the excessive smartphone usage children and moderate smartphone usage children. Excessive smartphone usage children would only exhibit pro-environmental behavior through social norms, whereas moderate smartphone usage children displayed higher personal norms and perceived behavioral control toward more environmentally friendly behaviors.

Supplemental Information

Supplemental Information 1 Raw data of the two control groups investigated, the excessive smartphone usage group accounted for 42.7% ( n = 96), whereas the moderate smartphone usage group was 57.3% ( n = 129)

This study specifically sought to measure three key dimensions: (1) perceived behavioral control, (2) personal norms, and (3) social norms that were considered influential to pro-environmental behavior. This research adopted the self-administered questionnaire survey method, which comprised of background and psychological variables developed according to the three key dimensions identified earlier. Overall results showed that both males (48.9%, n = 110) and females (51.1%, n = 115) were relatively well represented in this study.

Click here for additional data file.

We would like to thank the principals and teachers in the participating schools, as well as the children and their parents who have contributed to the successful completion of the survey. We are also very grateful to the anonymous reviewers for their constructive feedback that have contributed to the improvement of the manuscript.

Appendix A1. Survey Questionnaire 20170810Ver2

Additional Information and Declarations

Competing Interests

Author Contributions

Ethics

Data Availability

The authors declare there are no competing interests.

Wei-Ta Fang conceived and designed the experiments, performed the experiments, analyzed the data, authored or reviewed drafts of the paper, and approved the final draft.

Eric Ng conceived and designed the experiments, authored or reviewed drafts of the paper, and approved the final draft.

Shu-Mei Liu performed the experiments, analyzed the data, prepared figures and/or tables, authored or reviewed drafts of the paper, and approved the final draft.

Yi-Te Chiang analyzed the data, prepared figures and/or tables, and approved the final draft.

Mei-Chuan Chang conceived and designed the experiments, performed the experiments, authored or reviewed drafts of the paper, and approved the final draft.

The following information was supplied relating to ethical approvals (i.e., approving body and any reference numbers):

This study was approved by the National Taiwan Normal University Research Ethics Committee research and ethics committee (201707HS001) and agreed with active informed consent by the class teachers and school principals with parents having the option to opt their child out of the study.

The following information was supplied regarding data availability:

The raw measurements are available in the Supplemental File.

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
