# Peer review of "Determinants of pro-environmental behavior among excessive smartphone usage children and moderate smartphone usage children in Taiwan"

_PeerJ, doi:10.7717/peerj.11635_

## Round 0.1 · original submission · Major Revisions

Sorry for the delay, which is mainly due to the fact that I have had many difficulties in finding reviewers for your paper. I have now received one complete review, which is very clear. Please notice that the reviewer is against the publishability of your current work. However, also because of the interest in the topic, I have decided to let the door open for resubmission, but you have to convince the reviewer and myself.

I invite you to address the reviewer's point. You do not necessarily need to agree with all of them; in case you don't, please clarify why. I think that you should work extensively on the theoretical and hypothesis part, rendering the literature overview and the theory more tightly linked to the hypothesis you advance.

Looking forward to your revised article,

Reviewer 1 ·

Basic reporting

This paper intends to explore whether the personal norm and social norm can directly affect pro-environmental behavior or indirectly affect by perceived behavioral control between two types of children. They were divided into two groups according to their daily smartphone usage. The topic of this paper is interesting, but I also find some needs to improve.

1. The authors have better make a comprehensive examination and revision of the language of the manuscript. In line 67-69, the authors stated that “Studies show that children have abandoned physical outdoor activities and their curiosity about the natural environment, which may have direct health and behavioral benefits, and instead spending their time and energy on smartphones” . In this sentence, “and” is used to connect two sentences, but “spending” is not the predicate of the last sentence. And the clause introduced by “which” is also difficult to understand. The expression of this sentence would better be modified. Similarly, in line 75-77, the authors stated that “However, there are very few studies attempted to combine the two fields of smartphone usage and environmental behaviors together, and explore their relationship”. "And" can be replaced by "let alone", which may be more logical. And in line 77, “investigating specifically” may be better to be replaced by “specifically investigating”. In line 93-94, the authors stated that “ On the other hand, external attributions are causes of behavior result from some situations or events outside a person’s control.” This sentence concludes two predicates and it is incorrect.

2. In line 87, authors mentioned that “the two key elements involve are: (1) perceived behavioral control, and (2) personal norms.” I hope authors can provide relevant literature.

3. In line 94-96, the authors mentioned that social norms can be classified into two types. I did not figure out the purpose of this sentence, because its connection to the rest of this paper is not clear.

4. Importantly, the attribution theory is the framework of this study, but it does not play a prominent role in the construction of variable relationships. And the authors also did not clearly and sufficiently explain the relationship between key variables of this research. Although authors cited literature behind each hypotheses, the specific relationship and internal logic of each key variable remains unclear. It may be difficult for readers to understand why authors intend to construct that model. Authors should provide more details. For example, both personal norms and perceived behavioral control are key elements of internal attribution. Why do authors suggest PN predict PBC rather than PBC predicted PN?

5. Pro-environmental behavior is a key variable in this research, but the authors do not provide its definition in manuscript.

6. The authors propose ten hypotheses. It can be simplified. For example, the authors could propose that perceived behavioral control plays a mediating role between personal norms and pro-environmental behavior, rather than propose two separate hypotheses. And the authors could discuss more about why they propose these hypotheses.

Experimental design

1. This research basically confroms with technical & ethical standards. But I noticed that the authors did not mention whether the subjects were paid after completing the questionnaires in line142-162.

2. Line 199-200, the authors stated that “A five-point Likert scale (i.e., 1 = “Strongly disagree” to 5 = “Strongly agree”) was applied for the measurement in this study.” Does each scale have a five-point score?

3. It is clear and simple to understand the aim of this paper. However, as I mentioned before, the key variable of pro-environmental behavior was not well defined in the manuscript. And the authors did not clearly state the internal logic relationship between the key variables of this research. That has influenced the reliability and meaning of the submission.

Validity of the findings

In Results part, the authors clearly reported their results in detail and it is a strength of this submission. but I also find some small mistakes.

1. In line 287-289, authors stated “In the excessive smartphone usage group (as shown in Table 7), all of the convergent validity, i.e. average variance extracted (AVE) values were above 0.50, and likewise the composite reliability (CR) values were at least 0.70.” “all of the convergent validity” does not have corresponding predicate and object which could lead to difficulty in reading. The same situation exists on line319-321.

2. In line 297, the authors stated that "the dimensions affected were...". It would be better to be replaced by "affected dimensions". And the authors adopted the last expression in line 327. In addition, the ";" in line 327 is redundant.

3. There are some highly repetitive statements in Results part and it can be simplified (i.e. line 303-313)

Moreover, after reading the Discussion part, I also have some question.

1. Importantly, I merely find references to support authors' discussion which is harmful to the credibility of this research. For example, in line 406-409, the authors intend to explain the mediating role of perceived control between personal norms and pro-environmental behavior by self-resilience. But authors do not provide references or elaborate on the role of self-resilience, which makes the explanation become faint and powerless. It is necessary to cite related literature to enhance the persuasiveness of the discussion part.

2. In line 364, authors mentioned that “Results suggested that the causal relationship”. It would be better not to write in such an absolute way, because the causal relationship is only a theoretical deduction.

3. In line 377, the authors mentioned that “in countries such as China, Korea, Japan, Singapore and Taiwan”. Taiwan is not an independent country both in China and international, so the expression should be revised.

Additional comments

This research focuses on an interesting topic, aiming to figure out the relationship and internal mechanism between smartphone usage and pro-environmental behavior. The authors have cited many previous literature in their Introduction part, but they do not analyse the relation between their own research and previous studies. In addition, the hypotheses' theoretical basis and logical foundation was faint and the authors should make revision to improve their credibility.

---

## Round 0.2 · Minor Revisions

I have carefully read your article and found it improved and almost ready for acceptance. However, I invite you to polish English (for example, the word evidence is singular, not plural).

---

## Round 0.3 · accepted · Accept

I am happy to inform you that your paper has been accepted for publication on PeerJ.